# Engineered human meniscus' matrix-forming phenotype is unaffected by low strain dynamic compression under hypoxic conditions

**Alexander R. A. Szojka, Colleen N. Moore, Yan Liang, Stephen H. J. Andrews, Melanie Kunze, Aillette Mulet-Sierra, Nadr M. Jomha, Adetola B. Adesida**⊙*

Divisions of Orthopaedic Surgery and Surgical Research, Department of Surgery, Faculty of Medicine & Dentistry, University of Alberta, Edmonton, Alberta, Canada

* adesida@ualberta.ca

**Data Availability Statement:** Data are available from http://dx.doi.org/10.17632/4ttbwnsf6z.1.

## Abstract

Low oxygen and mechanical loading may play roles in regulating the fibrocartilaginous phenotype of the human inner meniscus, but their combination in engineered tissues remains unstudied. Here, we investigated how continuous low oxygen ("hypoxia") combined with dynamic compression would affect the fibrocartilaginous "inner meniscus-like" matrix-forming phenotype of human meniscus fibrochondrocytes (MFCs) in a porous type I collagen scaffold. Freshly-seeded MFC scaffolds were cultured for 4 weeks in either 3 or 20% $O_2$ or pre-cultured for 2 weeks in 3% $O_2$ and then dynamically compressed for 2 weeks (10% strain, 1 Hz, 1 h/day, 5 days/week), all with or without TGF-β3 supplementation. TGF-β3 supplementation was found necessary to induce matrix formation by MFCs in the collagen scaffold regardless of oxygen tension and application of the dynamic compression loading regime. Neither hypoxia under static culture nor hypoxia combined with dynamic compression had significant effects on expression of specific protein and mRNA markers for the fibrocartilaginous matrix-forming phenotype. Mechanical properties significantly increased over the two-week loading period but were not different between static and dynamic-loaded tissues after the loading period. These findings indicate that 3% $O_2$ applied immediately after scaffold seeding and dynamic compression to 10% strain do not affect the fibrocartilaginous matrix-forming phenotype of human MFCs in this type I collagen scaffold. It is possible that a delayed hypoxia treatment and an optimized pre-culture period and loading regime combination would have led to different outcomes.

## Introduction

The menisci play important mechanical roles in the knee [1, 2]. In humans, their weight-bearing inner region develops a fibrocartilaginous phenotype throughout life with morphological differences from the outer region (Table 1) [3–9]. In adults, the inner region is avascular and

**Funding:** Financial or material grant support for this study was provided by: Natural Sciences and Engineering Research Council (NSERC RGPIN-2018-06290 Adesida); Canadian Institutes of Health Research (CIHR MOP 125921 Adesida); Edmonton Civic Employees Charitable Assistance Fund (ECECAF RES0036207, Adesida); Canada Foundation for Innovation (CFI 33786, Adesida); University Hospital of Alberta Foundation (UHF RES0028185, RES0045921 Adesida); Edmonton Orthopaedic Research Committee; Integra Lifesciences (USA) (in-kind donation of the type I collagen scaffold). Salary support for this study was provided by: ARAS: Alexander Graham Bell Scholarship, NSERC (nserc-crsng.gc.ca); President's Doctoral Prize of Distinction, University of Alberta (ualberta.ca); Canadian Institutes of Health Research; Queen Elizabeth II Scholarship program, Alberta Government; Alberta Graduate Scholarship, Student Aid Alberta; CNM: Alberta Innovates Health Solutions (AIHS) Summer Studentship; YL: Li Ka Shing Sino-Canadian Exchange Program; SHJA and MK: Alberta Cancer Foundation-Mickleborough Interfacial Biosciences Research Program (ACF-MIBRP 27128 Adesida); AMS: Canadian Institutes of Health Research (CIHR MOP 125921 Adesida); NMJ and ABA: University of Alberta. The funders had no role in study design, data collection and analysis, decision to publish, or preparation of the manuscript.

**Competing interests:** The authors have declared that no competing interests exist.

non-healing whereas the outer region has a peripheral blood supply that provides some healing capacity. Inner meniscus injuries disrupt knee function and may cause early osteoarthritis development [10–12].

The aim of meniscus tissue engineering is to generate meniscus tissue replacements that restore function and prevent osteoarthritis after non-healing injuries. Knowledge of how the human inner meniscus phenotype is regulated will facilitate development of these replacements. Human tissue engineering models allow study of how meniscus fibrochondrocytes (MFCs), the cells primarily responsible for synthesis and maintenance of the inner meniscus, regulate fibrocartilaginous matrix formation in response to biophysical factors present in the knee joint [13]. Two factors believed to play regulatory roles in the inner meniscus phenotype are oxygen tension and mechanical loading.

Oxygen appears to mediate the inner meniscus phenotype. The inner regions of the adult menisci exist in a low oxygen environment, as they are avascular and surrounded by the hypoxic synovial fluid [14–16]. Low oxygen conditions, described as "hypoxic" or "physioxic" to reflect the native inner meniscus microenvironment, affect development of meniscus cell-based engineered tissues [17–25]. Hypoxic culture of human meniscus cell-based engineered tissues promotes formation of cartilaginous matrix components type II collagen and aggrecan [18–20, 23, 24], which provide compression resistance through their water-retaining properties [26, 27]. Hypoxia-mediated chondrogenesis is regulated through transcription factor hypoxia-inducible factor-1 (HIF-1), gene expression of transcription factor SRY-related high-mobility group (HMG) box-9 (*SOX9*), and growth factors such as transforming growth factor-β (TGF-β) [18, 19, 24]. Studies in our research group, however, show that cartilaginous matrix formation under hypoxia by human meniscus cells differs between scaffold-free pellets and a larger-scale type I collagen scaffold, which warrants further investigation [21, 22, 24]. One possible explanation for these differing results is the cell source. The scaffold studies used meniscus from older (ages 54–79) donors undergoing knee replacements for severe osteoarthritis whereas the pellet study used meniscus tissues from younger (ages 19–46) donors without osteoarthritis. As well, the pellet study had continuous hypoxia treatment without re-oxygenation during medium changes using an XVivo incubator system (Biospherix, USA), whereas the scaffold studies used standard hypoxia incubators with tissues re-oxygenated briefly during medium changes.

Mechanical loading also appears to mediate the inner meniscus phenotype [3–6]. Dynamic deformational loading (e.g. dynamic tension, dynamic compression) and cyclic hydrostatic pressure [28], which mimics the pressurization of fluid in cartilaginous matrix during weight-bearing, have been applied to study the biosynthetic response of meniscus explants and meniscus cell-based engineered tissues [29–46]. Only five mechanical loading studies to date use human meniscus cells or tissues. These studies showed either improved cartilaginous matrix

**Table 1. Summary of relevant general inner-outer meniscus differences in adults.**

| | Inner (2/3 or more) | Outer (1/3 or less) |
|---|---|---|
| **Tissue phenotype** | Fibrocartilage | Dense fibrous connective tissue (ligament and tendon-like) |
| **Cell morphology** | Fibrochondrocytes (MFCs): a mixed population of fibroblast and chondrocyte-like cells | Fibroblast-like |
| **ECM composition** | Type I collagen, trace amounts of type II collagen and aggrecan | Type I collagen |
| **Blood supply and healing capacity** | Avascular, non-healing | Vascularized, healing |
| **Loading environment** | Weight-bearing (compression), tension | Tension |

formation under loading or no effects [29–33]. Although dynamic compression has been applied to investigate the mechanical behaviour of human meniscus explants (see, for example [47]), its effects on biosynthesis in human meniscus explants and engineered tissues remain unstudied.

Hypoxia and mechanical loading have not previously been combined for engineering meniscus cell-based tissues *in vitro*. Confined dynamic compression under hypoxia of bovine chondrocytes in polyurethane scaffolds seemed to synergistically stabilize the chondrocyte phenotype in a preliminary study [48]. TGF-β3 supplementation and hypoxia but not unconfined dynamic compression had beneficial effects on chondrogenic differentiation in porcine bone marrow-derived mesenchymal stem cells (BM-MSCs) in agarose [49]. Dynamic compression of rabbit BM-MSCs in agarose under normal oxygen levels, however, induced chondrogenesis as effectively as TGF-β1 supplementation [50].

Here, we investigated how combined hypoxia and dynamic loading may regulate the fibrocartilaginous inner meniscus phenotype, which could lead to a "biomimetic" strategy for meniscus tissue engineering [21, 24, 51]. We used human MFCs from the inner meniscus regions, isolated from young donors undergoing partial meniscectomy but without osteoarthritis, because they are a promising cell source for clinical translation. The MFCs were seeded on a porous biomaterial scaffold made of type I collagen, the main biochemical constituent of human inner meniscus. We first investigated hypoxia effects under static conditions and then dynamic compression effects under hypoxic conditions. The experiments were performed with or without TGF-β3 supplementation to determine if hypoxia under static conditions and hypoxia combined with dynamic compression could induce a fibrocartilaginous matrix-forming response without exogenous growth factors.

Our hypotheses were that the fibrocartilaginous, "inner meniscus-like" matrix-forming phenotype of MFCs would be: i) promoted by hypoxia but dependent upon TGF-β3 supplementation [24], and ii) promoted by dynamic compression under hypoxia regardless of TGF-β3 supplementation.

## Materials and methods

Two related experiments—Part 1: hypoxia and TGF-β3 supplementation effects, and Part 2: dynamic compression and TGF-β3 supplementation effects under hypoxia—were carried out as outlined in Fig 1. Unless otherwise specified all methods have been previously described in detail [21, 23].

### Ethics and sample collection

Meniscus specimens were collected during arthroscopic partial meniscectomies at the Grey Nuns Community Hospital and the University of Alberta Hospital in Edmonton with patient consent waived in accordance with the University of Alberta human research ethics board's approval #Pro00018778 to use surgical cast-offs with non-identifying information for scientific research purposes.

### Cell and tissue culture

Meniscus tissues from seven male donors ages 19–39 were digested overnight in type II collagenase (0.15% w/v; 300 units/mg). Cells from all donors were kept separate from one another, i.e. cells from different donors were never pooled to preserve inter-donor variation (S1 Dataset). Four donors were cultured for part 1 only, and three were cultured for both parts 1 and 2. The exact breakdown of data generated from each donor is available in S1 Dataset. As a recovery period, MFCs were placed in standard tissue culture flasks in monolayer with a standard

① Isolate and expand human meniscus fibrochondrocytes (MFCs) to passage 2 (P2) with TGF-β1 and FGF-2

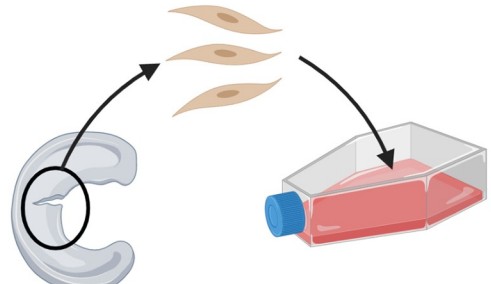

② Seed MFCs onto porous cylindrical type I collagen scaffolds using a density of $5 \times 10^6/cm^3$

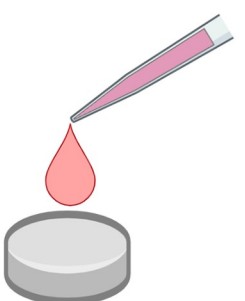

③a Part 1: Culture the cell-seeded scaffolds for 4 weeks in medium with or without TGF-β3

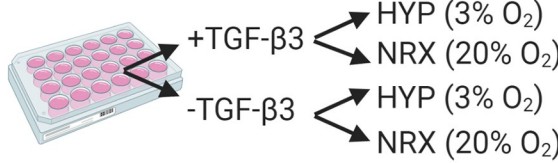

+TGF-β3 → HYP (3% $O_2$), NRX (20% $O_2$)

-TGF-β3 → HYP (3% $O_2$), NRX (20% $O_2$)

③b Part 2: After 2 weeks pre-culture, apply dynamic compression (DC) 5x/week for 2 weeks all in HYP in medium with or without TGF-β3

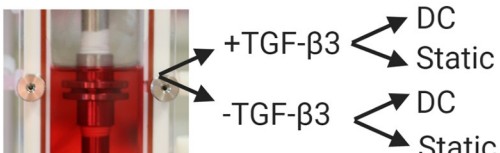

+TGF-β3 → DC, Static

-TGF-β3 → DC, Static

④ Evaluate tissue properties by mechanical testing, biochemistry, histology, and PCR

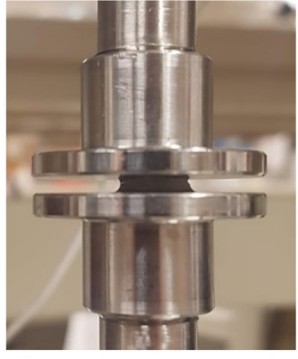
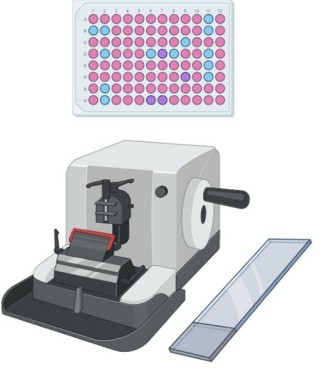
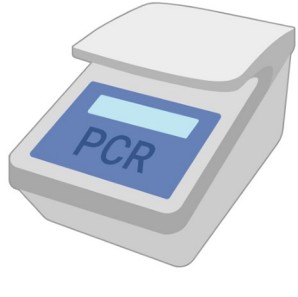

**Fig 1. Study outline.** HYP: Hypoxia. NRX: Normoxia.

basal expansion medium containing 10% v/v fetal bovine serum (FBS) for 48 hours. For cell expansion, MFCs were put into fresh flasks at a density of $10^4/cm^2$ in humidified incubators with 5% $CO_2$. The basal medium (BM) was supplemented with 1 ng/mL TGF-β1 and 5 ng/mL FGF-2 in the main experiments, a combination that was shown to increase proliferation rates while maintaining the cells' matrix-forming capacity compared to basal medium (BM) and BM+FGF-2 alone (S1 Fig).

MFCs were expanded to passage 1 or 2 (population doublings 2.9–4.1), a degree of expansion shown previously to produce MFCs with good fibrocartilaginous matrix-forming capacity [23]. MFCs were collected and seeded onto 10-mm diameter type I collagen scaffolds (3.5-mm thick, pore size 115±20 μm) at a density of $5×10^6/cm^3$. For tissue formation, cell-seeded scaffolds were cultured in a defined serum-free chondrogenic tissue culture medium, but with or without the standard addition of 10 ng/mL of TGF-β3 in incubators containing 3 or 20% $O_2$ and 5% $CO_2$ (X3 XVivo, Biospherix, USA) for up to 4 weeks [23, 24]. Medium changes for hypoxia groups occurred within a 3% $O_2$ environment, i.e., the hypoxia treatment was continuous.

## Dynamic compression

After two weeks of pre-culture with or without TGF-β3 supplementation in hypoxia (3% $O_2$) engineered tissues were transferred onto custom platens fitted to chambers containing 110 mL of the tissue culture medium on a Biodynamic 5210 system (TA Instruments, USA). Tissues were pre-loaded to 0.01-N each and dynamically compressed at least five times per week for two more weeks to 10% strain at 1 Hz for 1 hour per day with static vehicle controls in replicate chambers. Apart from reduced dynamic loading time (2 weeks vs. 3 & 6 weeks), this regime is the same as was applied in the context of comparing individual and combined effects of hypoxia (5% $O_2$), dynamic compression, and TGF-β3 supplementation on chondrogenic differentiation with or without a 3 week pre-culture period, but in a different culture model (four month old porcine bone marrow-derived mesenchymal stem cells in 2% agarose hydrogels) [49].

## Mechanical analysis

During loading, the compression modulus from the first and last cycles were determined by dividing the change in force by the applied strain and the cross-sectional area. After the culture period, tissues were individually tested using unconfined compression stress relaxation tests. Tissues were pre-loaded to 0.01 N and then compressed to 10% strain using a strain rate 50% strain/s [52, 53]. Tissues were allowed to relax to equilibrium. The peak change in force was used to calculate the instantaneous modulus and the equilibrium change for the equilibrium modulus.

## Quantitative biochemical and qualitative histological and immunofluorescence analysis

For biochemistry, after mechanical testing tissues were solubilized in proteinase K. Glycosaminoglycan contents were measured using the dimethylmethylene blue (DMMB) assay and DNA contents with the CyQUANT cell proliferation assay.

For qualitative histology, tissues were processed for paraffin embedding, sectioned at 5 μm, and stained with all of Safranin-O, Fast Green FCF, and Meyer's haematoxylin. Immunofluorescence was performed for types I and II collagens and cell nuclei by 4',6-diamidino-2-phenylindole (DAPI). The anti human type I collagen antibody showed only minute traces of cross reactivity with the bovine type I collagen scaffold.

## Gene expression analysis

RNA was isolated using TRIzol reagent and reverse transcribed into cDNA for quantitative real-time polymerase chain reaction (qRT-PCR) analysis using the gene specific primers listed in S1 Table. Expression of genes of interest was normalized to the mean expression level of reference genes YWHAZ, β-actin, and B2M and presented using the $2^{\Delta Ct}$ method [54]. Statistics were computed using ΔCt values.

## Statistical analysis

Mean replicate values for each donor were used for statistical analysis in SPSS 26 (IBM, USA). Two-way ANOVA was performed with TGF-β3 supplementation and oxygen tension or dynamic compression as fixed factors and donor as a random factor. All data was assumed to be normally distributed. When the interaction effect between fixed factors was significant, four pairwise comparisons were made (e.g. 1. NRX/-TGF-β3 vs. HYP/-TGF-β3; 2. NRX/+TGF-β3 vs. HYP/+TGF-β3; 3. NRX/-TGF-β3 vs. NRX/+TGF-β3; 4. HYP/-TGF-β3 vs. HYP/+TGF-β3). When the fixed factor interaction effect was not significant, the significance of the main effects of each fixed factor were assessed.

Linear correlations (Fig 5E–5G) were computed after mean centering data from each donor to reduce the influence of variability between donors. This involved dividing values by the overall donor mean and multiplying by the grand mean.

## Results

In preliminary work, FGF-2 supplementation increased proliferation rates compared to the basal medium (1.1-fold, p = 0.030) (Panel A in S1 Fig). FGF-2 combined with TGF-β1 (T1F2) caused a larger increase (1.6-fold, p<0.001) (Panel A in S1 Fig). GAG/DNA and mechanical properties were comparable across conditions, indicating that the matrix-forming capacity of MFCs was maintained (Panels B–E in S1 Fig). Given these results, the T1F2 expansion strategy was used throughout the rest of the study.

In Part 1 (static conditions), TGF-β3 supplementation significantly increased fibrocartilaginous, "inner meniscus-like" matrix formation as assessed by GAG, GAG/DNA, and mechanical properties (Fig 2A, 2C, 2D and 2E). DNA contents in TGF-β3 supplementation groups were also significantly higher (Fig 2B).

Hypoxia did not affect GAG nor GAG/DNA in TGF-β3 supplementation groups (Fig 2A and 2C). Constructs overall had 15% less contraction in hypoxia than normoxia (Fig 2F). During mechanical testing, the instantaneous force change across the strain step was not different between hypoxia and normoxia in TGF-β3 supplementation groups (p = 0.85, S1 Dataset). However, when the instantaneous modulus was computed by dividing the force change by the cross-sectional area it was 20% lower in hypoxia (p<0.05) (Fig 2D). Construct wet weights are available in S2 Fig.

Histologically, groups without TGF-β3 supplementation had little synthesized matrix (Fig 3). While groups with TGF-β3 supplementation had regions of abundant synthesized matrix and stained more intensely with Fast Green than -TGF-β3 groups, much of the scaffold showed no cell activity (Fig 3). Matrix staining positively for Safranin-O, indicating a fibrocartilaginous phenotype, was detected only in the TGF-β3 supplementation groups (Fig 3). There were no large and consistent hypoxia effects assessed by histology alone; however, taken together with the quantitative results for DNA content and % diameter contraction, there was evidence for increased cell density in NRX/+TGF-β3 compared to HYP/+TGF-β3 (Figs 2B, 2F and 3). TGF-β3 supplementation groups had increased type I collagen intensity and were the

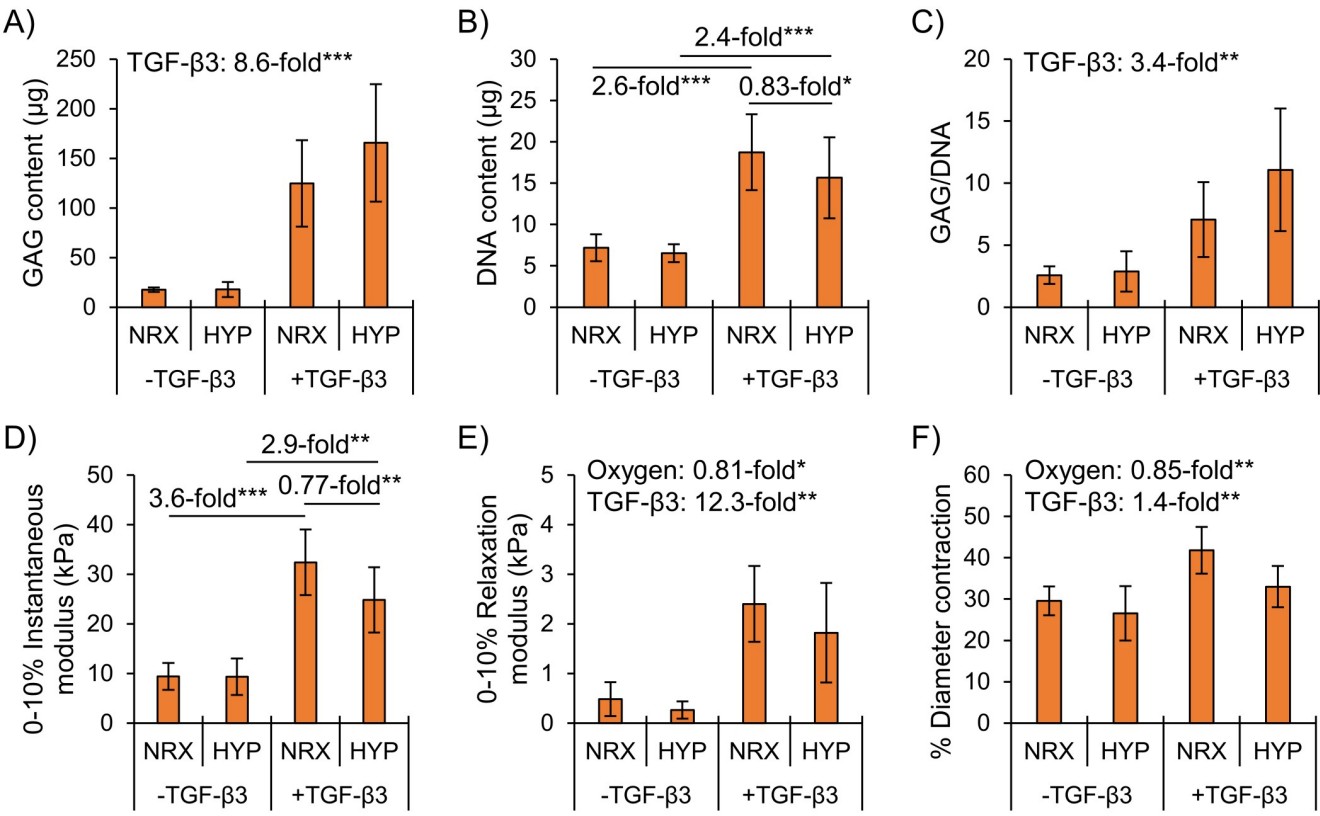

**Fig 2. Biochemical accumulation was superior in TGF-β3 supplementation groups and mechanical properties were superior in normoxia.** Differences were assessed by two-way ANOVA. *: p<0.05, **: p<0.01, ***: p<0.001. NRX: Normoxia, 20% $O_2$. HYP: Hypoxia, 3% $O_2$.

only ones to show the presence of cartilaginous matrix component type II collagen, albeit only in traces (Panel A in S3 Fig).

TGF-β3 supplementation increased fibrocartilaginous matrix-related mRNA expression, whereas hypoxia had no detectable effects (Fig 4). Taken with protein-level observations, this indicates that normoxia-cultured constructs had superior mechanical properties due to different contractile characteristics rather than differences in functional matrix. Hypoxia thus did

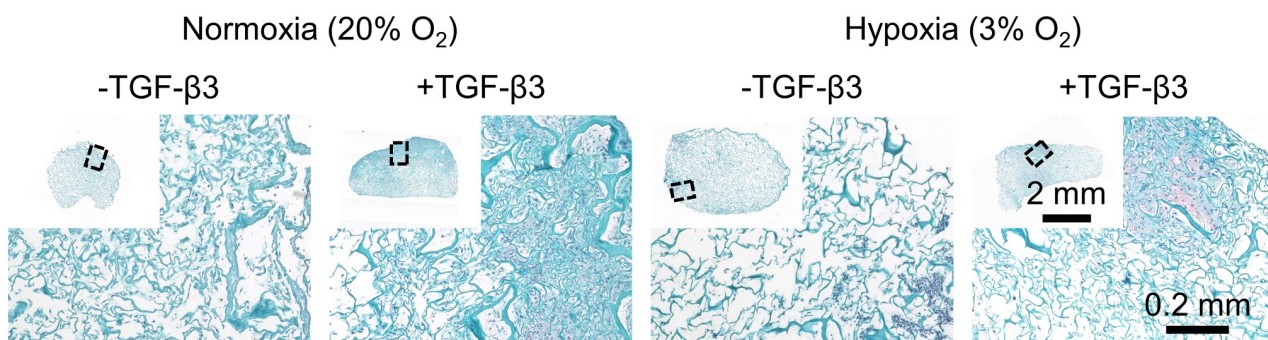

**Fig 3. Safranin-O staining was superior in TGF-β3 supplementation groups and was not affected by hypoxia.** Staining for a representative donor is presented. Staining was performed by combined Safranin-O (sulphated proteoglycans, pink), Fast Green (proteins, turquoise), and haematoxylin (cell nuclei, purple-dark green here).

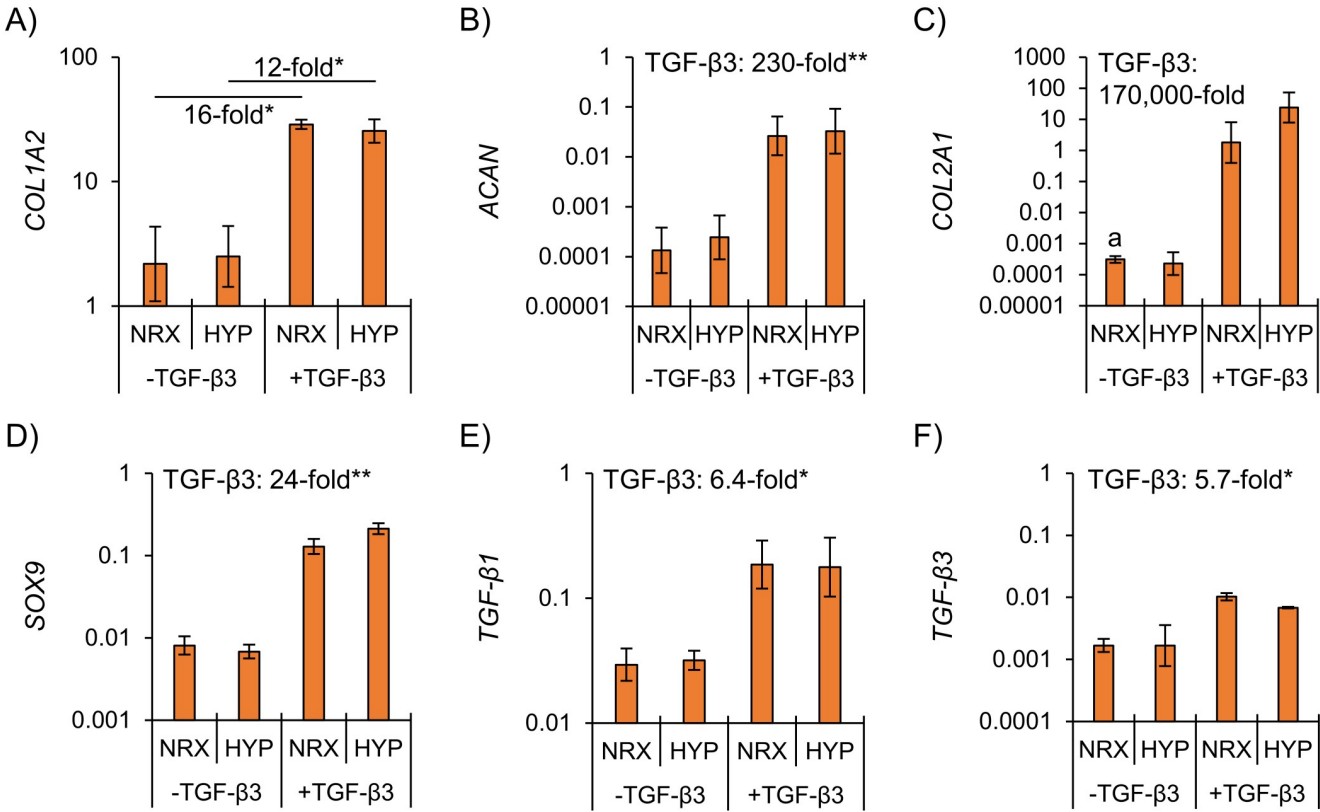

**Fig 4. Inner meniscus gene expression was enhanced by TGF-β3 supplementation and was not affected by hypoxia.** Differences were assessed by two-way ANOVA. *: p<0.05, **: p<0.01. NRX: Normoxia, 20% $O_2$. HYP: Hypoxia, 3% $O_2$. a: No amplification within 40 PCR cycles for at least one donor.

not induce a fibrocartilaginous matrix-forming response regardless of TGF-β3 supplementation.

In Part 2, stress relaxation during the loading period could only be observed in +TGF-β3 groups (Fig 5A–5C). The forces during loading were lower in groups without TGF-β3 supplementation, though they were higher than those due to friction alone in the bioreactor (Fig 5D). This confirmed that the tissues in -TGF-β3 groups were experiencing dynamic compression despite their poor mechanical properties. Regardless of TGF-β3 supplementation, platen liftoff occurred during the decompression phase near the end of the loading period, with forces dropping to 0 N below ~5% strain (Fig 5D). However, during the compression phase (upper part of curves) the forces showed an immediate rise (Fig 5D). This indicated that contact was quickly re-established and that the elastic rebound of the tissues was only slightly too slow for the applied waveform (Fig 5D). The tissues cultured with TGF-β3 supplementation had similar heights from day to day (Fig 5E). The initial and final modulus during loading periods, calculated using the changes in force across cycles at the start and end of the loading periods, showed clear increases over time in +TGF-β3 groups only (Fig 5F & 5G). After the two-week dynamic compression period, there were no differences in mechanical properties between loaded and unloaded tissues (Fig 6D & 6E). Thus, the rise in modulus during loading was attributed to increased culture time rather than to dynamic compression exerting matrix-strengthening effects.

Without TGF-β3 supplementation, biochemical and mechanical properties remained low despite the "biomimetic" combination of dynamic compression and hypoxia

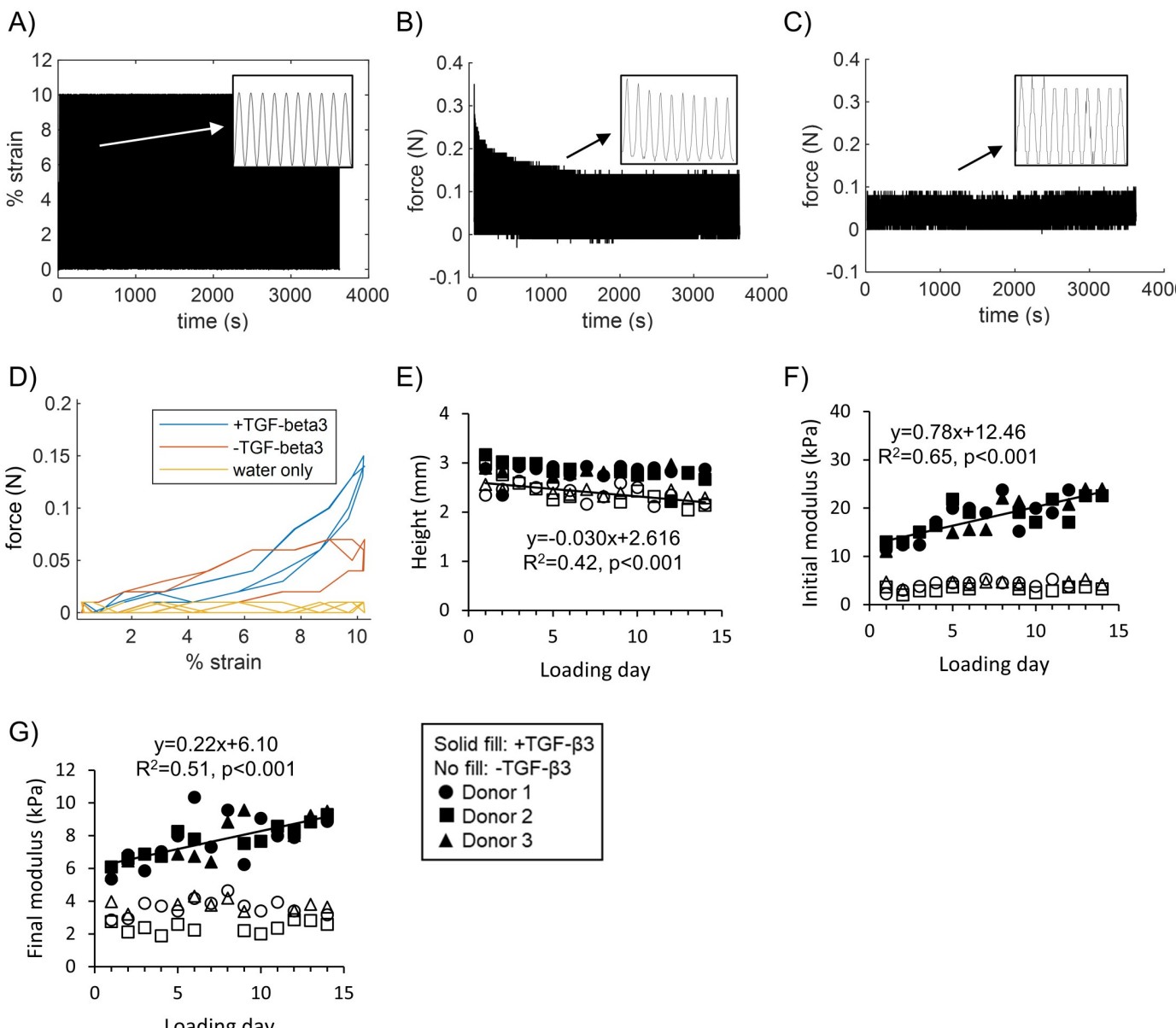

**Fig 5. Dynamic compression loading summary.** A) Tissues were compressed from 0–10% strain at 1 Hz for one hour (3600 cycles). B & C) Representative force vs. time plots for tissues (loading day 10) that had medium supplemented with or without TGF-β3, respectively. D) Representative force vs. strain plots for two cycles near the end of a loading period (loading day 10). The water only group was included to show the magnitude of any frictional forces using a comparable platen stroke. E) Sample heights were measured on each loading day and were used to calculate displacements for application of the 10% strain waveform. F & G) Peak and equilibrium moduli calculated using the force changes across the first and last cycles, respectively, of each loading period for each donor. Only significant (p<0.05) linear correlations were included.

(Fig 6A, 6C, 6D and 6E). TGF-β3 supplementation increased all measured properties, but there were no consistent effects of dynamic compression (Fig 6).

Safranin-O staining was detected only in TGF-β3 supplementation groups, with matrix being most concentrated at the outer edges (Fig 7). Dynamic compression had no discernable effects by histology (Fig 7). As in part 1, immunofluorescence confirmed the universal presence of type I collagen (Panel B in S3 Fig). TGF-β3 supplementation groups had increased type I collagen intensity and were the only ones to show the presence of type II collagen (S3B Fig).

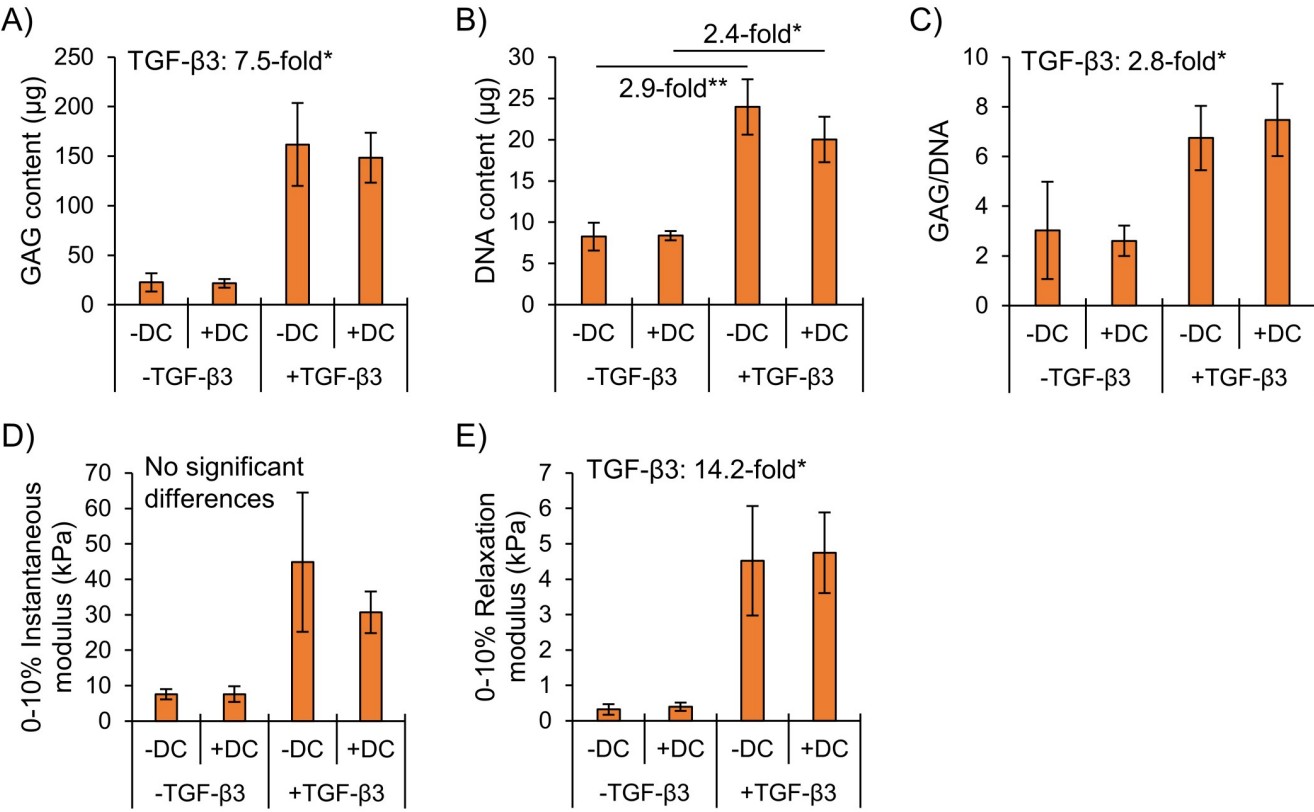

**Fig 6. Biochemical and mechanical properties were not affected by dynamic compression under hypoxia, regardless of TGF-β3 supplementation.** Tissues were pre-cultured statically for two weeks followed by two weeks of static or dynamic culture (10% strain, 1 Hz, 1h/day, 5×/week for two weeks). Differences were assessed by two-way ANOVA. *: $p < 0.05$, **: $p < 0.01$. DC: Dynamic compression.

Dynamic compression had no consistent effect on expression of matrix-related genes (Fig 8). The only significant dynamic compression effect at the mRNA level was a decrease in *TGF-β1* (0.86-fold in DC vs. static, $p < 0.01$) (Fig 8F).

Given that hypoxia under static conditions and dynamic compression under hypoxia did not affect the fibrocartilaginous matrix-forming phenotype, established hypoxia and

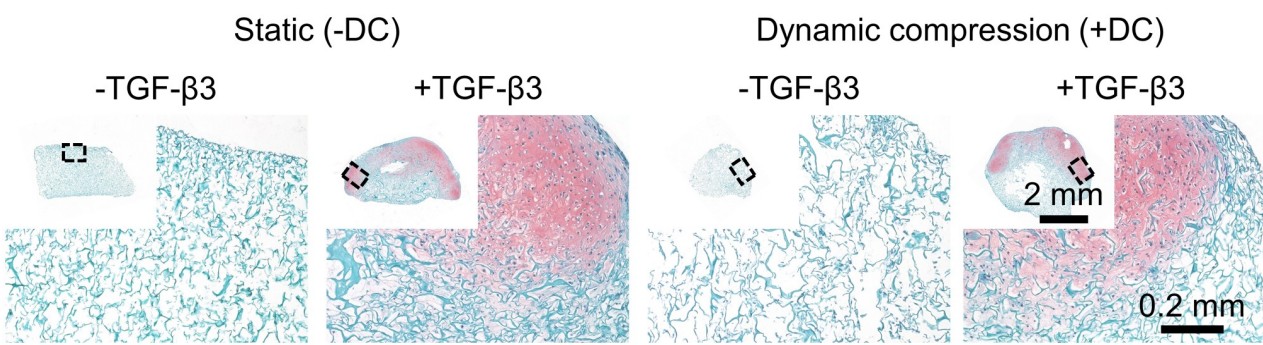

**Fig 7. Safranin-O staining was not affected by dynamic compression under hypoxia, regardless of TGF-β3 supplementation.** Staining was performed by combined Safranin-O (sulphated proteoglycans, pink), Fast Green (proteins, turquoise), and haematoxylin (cell nuclei, purple-dark green here).

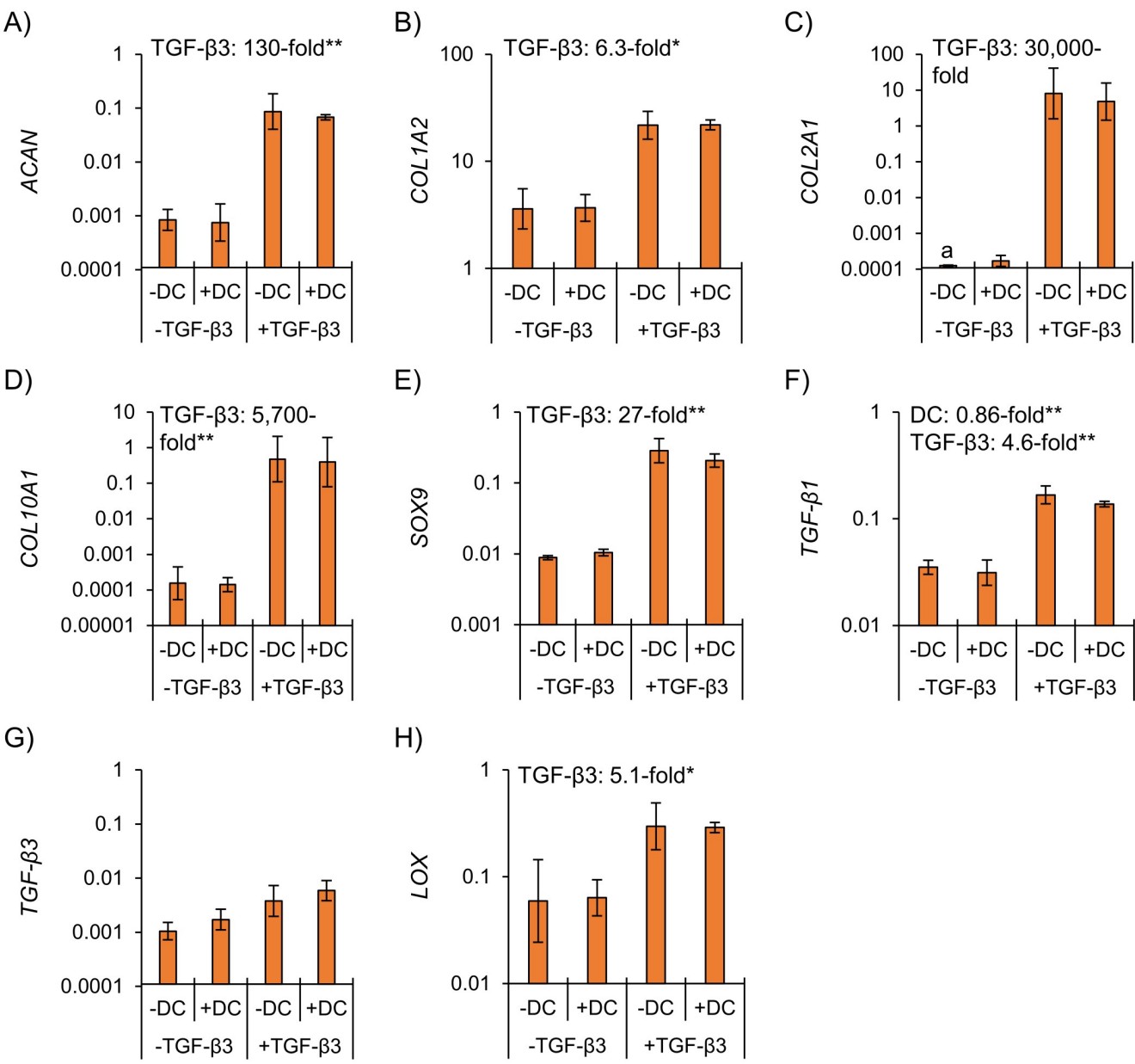

**Fig 8. Inner meniscus gene expression was minimally affected by dynamic compression under hypoxia regardless of TGF-β3 supplementation.** Tissues were pre-cultured statically for two weeks followed by two weeks of static or dynamic culture (10% strain, 1 Hz, 1h/day, 5×/week for two weeks). Differences were assessed by two-way ANOVA. *: p<0.05, **: p<0.01. DC: Dynamic compression. a: No amplification within 40 PCR cycles for one donor.

loading-sensitive genes were measured to confirm that the cells were at least responsive to the treatments. Hypoxia-sensitive genes *LOX* and *VEGF* seemed higher in hypoxia but these were not statistically significant (Fig 9A & 9B). *HIF-1α* and *HIF-2α*, genes whose products regulate expression of hypoxia-sensitive genes including *VEGF* and *LOX* but themselves are not necessarily regulated by hypoxia at the gene level [55, 56], were expressed in all groups but not differentially (Fig 9C & 9D). Loading-sensitive genes *c-FOS* and *c-JUN* [57, 58] were not differentially expressed between groups (Fig 9E & 9F).

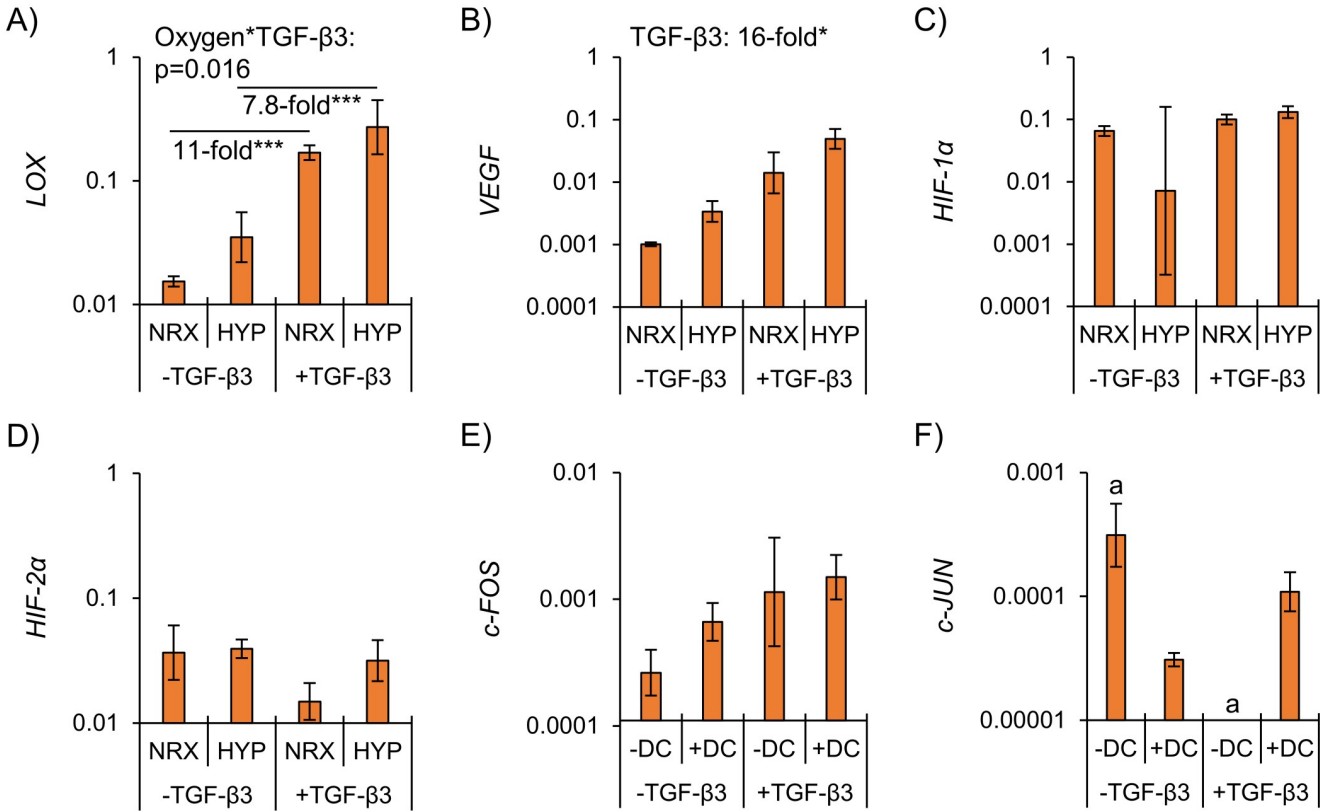

**Fig 9. The treatments were ineffective at regulating hypoxia (A–D) and mechanical loading (E & F)-related genes.** Differences were assessed by two-way ANOVA within donors (refer to the statistical analysis section). *: $p < 0.05$, ***: $p < 0.001$, a: No amplification within 40 PCR cycles for at least one donor. NRX: Normoxia, 20% $O_2$. HYP: Hypoxia, 3% $O_2$. DC: Dynamic compression.

## Discussion

In preliminary work, we showed that MFCs expanded with the basal medium (BM)+T1F2 had faster proliferation rates and comparable subsequent matrix-forming capacity to those expanded in BM alone and BM+FGF-2. The FGF-2-mediated proliferation rate increase compared to basal medium was lower than previously reported (e.g. 1.1-fold here vs. 1.5 fold before), which may reflect differences in flask confluency, serum batches, and meniscus specimen source: relatively young and healthy joints undergoing arthroscopic partial meniscectomy here vs. older and osteoarthritic joints undergoing knee replacements before [18, 23].

In part 1, we confirmed that continuous hypoxia alone was insufficient to induce a fibrocartilaginous matrix-forming phenotype in the type I collagen scaffold, complementing our result in the pellet model [24]. Unexpectedly, hypoxic compared to normoxic culture with TGF-β3 supplementation did not enhance the fibrocartilaginous matrix-forming phenotype. This contrasts with the synergistic increase in matrix formation by continuous hypoxia and TGF-β3 in the pellet model [24] and the hypoxia-mediated enhanced matrix formation in a polymeric scaffold layered with a biomimetic surface [20] by human meniscus-derived cells. However, this result is consistent with our previous work in this type I collagen scaffold using osteoarthritic joint meniscus cells, which had brief periods of re-oxygenation from hypoxia during medium changes [21, 22]. This indicates that the fibrocartilaginous matrix-forming response of meniscus-derived cells to TGF-β3 supplementation and hypoxia differs between pellets and this scaffold model and is similar in MFCs from partial meniscectomy and total knee

replacements, with re-oxygenation likely not playing a large role. We previously found differences in the MFC response to hypoxia in comparing models of a pellet and a meniscus decellularized matrix scaffold [25].

Possible underlying reasons for differences between studies may be oxygen diffusion dynamics, mediation of cell behaviour by the scaffold, cell density, and incubator oxygen tensions. Culture models such as pellets and self-assembling constructs may more accurately depict native tissues than traditional tissue engineering models incorporating an artificial scaffold [59]. It is possible that the differences in microenvironmental oxygen levels within the tissues were too small to observe a hypoxia-induced fibrocartilaginous matrix-forming phenotype. The increased contraction observed in normoxia would have reduced surface area for oxygen influx compared to the hypoxia group, reducing effective differences in oxygen levels. One strategy to mitigate confounding by differential contraction between oxygen tensions would be to pre-culture all tissues in a single oxygen condition to allow accumulation of contraction-resistant matrix before dividing tissues into different oxygen tensions. A pre-culture period in normoxia followed by exposure to hypoxic and mechanical loading conditions could mimic an aspect of human meniscus development: the menisci are initially fully vascularized but the inner regions become progressively devascularized in the first two decades of life with reducing oxygen supply and increasing mechanical loading [4, 15]. The hyaline cartilage aspect of the fibrocartilage phenotype develops in the inner regions during this time frame, suggesting this strategy could be used to possibly promote a more meniscus-like engineered tissue phenotype [4].

Given that mechanical loading seems to regulate the inner meniscus phenotype, we expected that dynamic compression would regulate MFC behaviour in the engineered tissues. Instead, dynamic compression under hypoxia did not induce a fibrocartilaginous matrix-forming phenotype without TGF-β3 supplementation nor enhance it in the presence of TGF-β3 supplementation. To confirm that cells within the tissues were responding to the loading treatment, we measured expression of loading-sensitive genes *c-FOS* and *c-JUN* and found no differences regardless of TGF-β3 supplementation [57, 58]. Data collected during loading showed that force changes across compression cycles always remained larger than friction even at the end of the 1 h loading period for both TGF-β3 groups. Although tissue level loading was applied, there was thus no detectable mechanotransduction response.

A plausible explanation is that the applied loading regime with small strains was too mild to consistently modulate cell behaviour and that the matrix distribution in the scaffolds was too inhomogeneous. A different result could plausibly have been obtained by applying a more aggressive loading regime and using a pre-culture strategy that promotes accumulation of more homogeneously-distributed matrix. The less stiff, matrix-scarce regions in the scaffolds would have borne more of the applied deformation than the stiff, matrix-rich regions. This would have shielded cells from the applied loading and diminished its effects. Two underlying reasons for the inhomogeneous cell distribution are the static seeding method, which is simple but may result in less uniform cell distributions than dynamic seeding strategies, and insufficient time for the scaffold to fill up with newly-synthesized matrix [60]. Increasing the scaffold seeding density and culture time are reliable means to improve matrix accumulation [61]. Our preliminary data showed that doubling the seeding density and increasing pre-culture time from 3 to 9 weeks caused a 36-fold increase in compression modulus, indicating the promise of these strategies for future investigations (S4 Fig).

The findings that TGF-β3 is necessary for matrix synthesis and that dynamic compression had few effects supports the results of work with porcine MSCs in agarose with a similar load regime [49]. However, unlike us, these authors observed hypoxia to exert chondrogenic effects and concluded that hypoxia is a more potent stimulator of chondrogenic differentiation than

dynamic compression [49]. The shared dynamic compression result may reflect the load regime being suboptimal for both models, while the differing oxygen effect may reflect the specific behaviour of MFCs on the type I collagen scaffold as discussed above [49]. Consistent with our loading results, a considerable proportion of studies applying dynamic compression in chondrocyte-based engineered tissues show no change in standard outcome measures of chondrogenesis [62]. In our opinion, this highlights the need for nuanced loading analysis and trial-and-error based optimization of loading regimes for each tissue culture model. A useful strategy based on our experiences here would be to reduce intra-donor variability, for example by using mean or pooled values from several replicates within donors and conditions. This would provide more stable values to aid in statistical detection of small mechanical loading effects with the same number of tissue donors. This could be useful because the added work from more tissue replicates is an order of magnitude smaller than that required for more donors in dynamic compression studies. A limitation of the commercial bioreactor used in the present study was its ability to only hold four replicates at once, which in this study needed to be divided amongst several assays (biochemistry, histology, gene expression). The inability to culture many replicates in this study could have precluded the detection of small dynamic compression effects with the limited number of cultured donors. Our future work will incorporate the use of a more optimal custom bioreactor.

Matrix staining for Safranin-O seemed more intense in Part 2 (dynamic compression and TGF-β3 supplementation effects) than in Part 1 (hypoxia and TGF-β3 supplementation effects) in +TGF-β3 supplementation groups. This was likely related to the different culture environments: part 1 used 24 well plates whereas part 2 used the commercial bioreactor, which had medium supplied in an extreme excess due to its large volume (27.5 mL/tissue construct/week compared to 3.7 mL/tissue construct/week in the 24 well plate).

Exogenous growth factors can overwhelm and mask the effects of stimuli such as dynamic compression [63]. In groups without TGF-β3 supplementation, cells bound to the scaffold with virtually no evidence of matrix formation after four weeks. However, this should not necessarily have prevented transduction of tissue-level deformations to cells [64]. Cells can directly interact with collagen scaffolds to allow transduction of tissue-level loads without substantial cell-synthesized matrix [64]. Yet, no loading effects were observed. It is possible that a more optimized loading regime may have led to a different outcome, such as one incorporating both shear and dynamic compression [65]. An alternative approach would be to withdraw growth factor supplementation after a pre-culture period. This would allow for accumulation of *de novo* synthesized matrix before applying dynamic compression. Withdrawal of TGF-β supplementation after a pre-culture period had remarkable chondrogenic effects in bovine chondrocytes and carpal bone marrow mesenchymal stem cells [66, 67].

## Conclusions

Our previous work demonstrated that: i) in pellets of human MFCs from partial meniscectomy, continuous hypoxia (3% incubator $O_2$) with TGF-β3 supplementation supports a more cartilage-like matrix-forming phenotype but hypoxia alone without growth factor supplementation (TGF-β3) is insufficient to induce matrix formation [24], and ii) in a type I collagen scaffold seeded with MFCs from total knee replacement (osteoarthritic joints), hypoxia with brief re-oxygenation during medium changes with TGF-β3 supplementation did not support a more cartilage-like matrix forming phenotype [21, 22]. The present study demonstrates that, in a type I collagen scaffold seeded with MFCs from partial meniscectomy, neither continuous hypoxia under static conditions nor hypoxia combined with dynamic compression to 10% strain induced a fibrocartilaginous inner meniscus-like matrix-forming phenotype in the

absence of TGF-β3 supplementation. They also did not promote the inner meniscus-like matrix-forming phenotype when it was induced by TGF-β3 supplementation. Different outcomes may have been obtained with a different timing of hypoxia application and if the pre-culture and dynamic loading regime parameters had been further optimized to incorporate, for example, larger strains. This study provides the first investigation of combined hypoxia and mechanical loading in meniscus cell-based engineered tissues and provides a baseline for future work. Eventually, this combinatorial strategy could be useful to better understand the behaviour of fibrochondrocytes in the native meniscus and to recapitulate differences between the inner and outer regions in tissue engineered meniscus replacements.

## Supporting information

**S1 Fig. TGF-β1 and FGF-2 increased proliferation rates and maintained MFCs' matrix-forming capacity in preliminary work.** A) All expansion medium contained 10% fetal bovine serum. B-D) Matrix formation was assessed after 3 weeks culture in this preliminary work. [TGF-β1]: 1 ng/mL, [FGF-2]: 5 ng/mL. Differences were assessed by one-way ANOVA. *: $p < 0.05$, **: $p < 0.01$, ***: $p < 0.001$.
(TIF)

**S2 Fig. Tissue wet weights were larger in HYP and were not affected by DC.**
(TIF)

**S3 Fig. Expression of types I/II collagens was increased by TGF-β3 supplementation and was not detectably affected by hypoxia nor DC.** Red: Type I collagen, green: Type II collagen, blue: Cell nuclei by DAPI. All images share a common scale bar.
(TIF)

**S4 Fig. Higher seeding density and longer culture times improved matrix formation and mechanical properties.** A) Tissue gross morphology at two seeding densities after 3- and 9-weeks' culture. B) Compression modulus in a ramp test up to 20% strain at 2% strain/minute. This mechanical test was modified from that used in the main study to better show the strain-stiffening behaviour of the tissue constructs.
(TIF)

**S1 Dataset. Quantitative data.**
(XLSX)

**S1 Table. Primer sequences used in qRT-PCR.** *: The COL1A2 primer described in 10.1089/ten.tea.2019.0306 is preferred for future work.
(DOCX)

## Acknowledgments

We acknowledge the contributions of Mr. Karamveer Lalh and Mr. Brayden D. Lyons to preliminary cell culture work.

## Author Contributions

**Conceptualization:** Alexander R. A. Szojka, Adetola B. Adesida.

**Data curation:** Alexander R. A. Szojka, Colleen N. Moore, Yan Liang, Melanie Kunze, Aillette Mulet-Sierra.

**Formal analysis:** Alexander R. A. Szojka, Stephen H. J. Andrews, Adetola B. Adesida.

**Funding acquisition:** Alexander R. A. Szojka, Nadr M. Jomha, Adetola B. Adesida.

**Investigation:** Alexander R. A. Szojka, Adetola B. Adesida.

**Methodology:** Alexander R. A. Szojka, Stephen H. J. Andrews, Melanie Kunze, Adetola B. Adesida.

**Project administration:** Alexander R. A. Szojka.

**Supervision:** Adetola B. Adesida.

**Writing – original draft:** Alexander R. A. Szojka.

**Writing – review & editing:** Alexander R. A. Szojka, Stephen H. J. Andrews, Nadr M. Jomha, Adetola B. Adesida.

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
