## [Decision Letter · Decision Letter 0]

18 Dec 2020

PONE-D-20-33847

Engineered human meniscus' matrix-forming phenotype is unaffected by low strain dynamic compression under hypoxic condition

PLOS ONE

Dear Dr. Adesida,

Thank you for submitting your manuscript to PLOS ONE. After careful consideration, we feel that it has merit but does not fully meet PLOS ONE’s publication criteria as it currently stands. Therefore, we invite you to submit a revised version of the manuscript that addresses the points raised during the review process.

We look forward to receiving your revised manuscript.

Kind regards,

Feng Zhao

Academic Editor

PLOS ONE

2.) Thank you for stating the following financial disclosure:

'No - The funders had no role in study design, data collection and analysis, decision to publish, or preparation of the manuscript.'

3.) We note that you have included the phrase “data not shown” in your manuscript. Unfortunately, this does not meet our data sharing requirements. PLOS does not permit references to inaccessible data. We require that authors provide all relevant data within the paper, Supporting Information files, or in an acceptable, public repository. Please add a citation to support this phrase or upload the data that corresponds with these findings to a stable repository (such as Figshare or Dryad) and provide and URLs, DOIs, or accession numbers that may be used to access these data. Or, if the data are not a core part of the research being presented in your study, we ask that you remove the phrase that refers to these data.

4.) Please include captions for your Supporting Information files at the end of your manuscript, and update any in-text citations to match accordingly. Please see our Supporting Information guidelines for more information: http://journals.plos.org/plosone/s/supporting-information.

**Comments to the Author**

1. Is the manuscript technically sound, and do the data support the conclusions?

Reviewer #1: Yes

Reviewer #2: Partly

2. Has the statistical analysis been performed appropriately and rigorously? 

Reviewer #1: Yes

Reviewer #2: N/A

3. Have the authors made all data underlying the findings in their manuscript fully available?

Reviewer #1: Yes

Reviewer #2: Yes

4. Is the manuscript presented in an intelligible fashion and written in standard English?

Reviewer #1: Yes

Reviewer #2: Yes

Reviewer #1: In this study, authors have evaluated the influence of hypoxia and dynamic compression on the fibrocartilaginous “inner meniscus-like” matrix-forming phenotype of human meniscus fibrochondrocytes (MFCs). Their results indicate that neither hypoxia under static culture nor hypoxia combined with dynamic compression had significant effects on expression of specific protein and mRNA markers for the fibrocartilaginous matrix forming phenotype. Moreover, hypoxia applied immediately after scaffold seeding and dynamic compression to 10% strain do not affect the fibrocartilaginous matrix-forming phenotype of human MFCs in this type I collagen scaffold. Although impressive and innovative, the authors are requested to address following issues in order to make the manuscript technically sound.

Major:

1. As authors mentioned, cells were isolated from 7 different donors with varied age. Cells from 4 donors were used in part I and cells from 3 donors were used in part II of experiments. Please clarify: In part I, did you mix the populations of cells isolated from various donors and then seeded into scaffolds or cells isolated from a particular donor were unmixed with the cells of other donors while seeding into scaffolds? It is important to clarify this fact since donor age significantly impacts the cellular performance.

2. Page 18, line 226: “There were no discernable hypoxia effects by histology (Fig 3)." - However, figure 3 indicates that normoxia samples had notably higher cellular proliferation than hypoxia as seen by densely packed nuclei in normoxia compared to hypoxia. - please clarify.

3. Please discuss- Is there a possibility if the order of hypoxia and normoxia exposure is altered, (i.e. after MFC seeding in the scaffold, cells are exposed to normoxia first and then hypoxia), it would benefit to achieve desired phenotypic properties?

Minor:

1. TGF-β- The “β” should be in italics. All Greek characters should be in italics.

2. Please explain: Why concentration of TGF-b3 supplementation was decided as 10 ng/mL?

Reviewer #2: This work appears to be an extension of their previous work. Lacks novelty. It cannot be considered a full paper, excluding the finding they already published, unless they thoroughly revise and expand their studies further. I have uploaded a separate comment sheet.

---

## [Author Response · Author response to Decision Letter 0]

9 Jan 2021

In addition to response below, please also refer to the Response to Reviewers attached file for the Figures mentioned below.

Reviewer #1:

In this study, authors have evaluated the influence of hypoxia and dynamic compression on the fibrocartilaginous “inner meniscus-like” matrix-forming phenotype of human meniscus fibrochondrocytes (MFCs). Their results indicate that neither hypoxia under static culture nor hypoxia combined with dynamic compression had significant effects on expression of specific protein and mRNA markers for the fibrocartilaginous matrix forming phenotype. Moreover, hypoxia applied immediately after scaffold seeding and dynamic compression to 10% strain do not affect the fibrocartilaginous matrix-forming phenotype of human MFCs in this type I collagen scaffold. Although impressive and innovative, the authors are requested to address following issues in order to make the manuscript technically sound.

We thank the reviewer for these comments and address their specific comments below.

Major:

1. As authors mentioned, cells were isolated from 7 different donors with varied age. Cells from 4 donors were used in part I and cells from 3 donors were used in part II of experiments. Please clarify: In part I, did you mix the populations of cells isolated from various donors and then seeded into scaffolds or cells isolated from a particular donor were unmixed with the cells of other donors while seeding into scaffolds? It is important to clarify this fact since donor age significantly impacts the cellular performance.

Thank you for bringing this up as it was not clear in the original manuscript. There was no pooling whatsoever of cells from different donors – the donors were all cultured separately to maintain their independence from one another.

This is now clarified in the manuscript in the Materials and methods, with reference to the supporting data which contains the complete quantitative data breakdown from each donor.

“Cells from all donors were kept separate from one another, i.e. cells from different donors were never pooled to preserve inter-donor variation (S1 Dataset).”

2. Page 18, line 226: “There were no discernable hypoxia effects by histology (Fig 3)." - However, figure 3 indicates that normoxia samples had notably higher cellular proliferation than hypoxia as seen by densely packed nuclei in normoxia compared to hypoxia. - please clarify.

Thank you for these observations. As far as detecting differences between groups, the qualitative histology as performed is good if differences are large and consistent between donors. The only ones clearly meeting these criteria were between +/-TGF-β3 groups. We are otherwise reluctant to make conclusions on the histological evidence alone because there is an uncomfortable risk of them being spurious due to variability between replicates and donors.

In response to the reviewer’s observations, we performed some more complete analysis by taking histology images together with the quantitative data for DNA content (an indicator for cell number) and scaffold contraction (reproduced below). When considered together, they suggest that the reviewer is likely correct in their observations. We note that since we do not have DNA content measured in the early phases of culture before proliferation could have taken place, we can only conclude that there was higher cell density in NRX vs. HYP, +TGF-β3 rather than specifically claiming more proliferation.

To reflect this, we have modified the related results sentence:

“There were no large and consistent hypoxia effects assessed by histology alone; however, taken together with the quantitative results for DNA content and % diameter contraction, there was increased cell density in NRX/+TGF-β3 compared to HYP/+TGF-β3 (Figs 2B, 2F, and 3).”

3. Please discuss- Is there a possibility if the order of hypoxia and normoxia exposure is altered, (i.e. after MFC seeding in the scaffold, cells are exposed to normoxia first and then hypoxia), it would benefit to achieve desired phenotypic properties?

The reviewer is correct - this was described in the third paragraph of the original discussion. In response to the comment, we now further discuss the value of such a strategy by linking it to an aspect of meniscus development below. We also provide very recent data below in our discussion with Reviewer 2 showing a 5.3-fold increase in SOX9 gene expression by a 5-day hypoxia treatment delayed until after a 3-week preculture period in normoxia.

“A pre-culture period in normoxia followed by exposure to hypoxic and mechanical loading conditions could mimic an aspect of human meniscus development: the menisci are initially fully vascularized but the inner regions become progressively devascularized in the first two decades of life with reducing oxygen supply and increasing mechanical loading[4,15]. The hyaline cartilage aspect of the fibrocartilage phenotype develops in the inner regions during this time frame, suggesting this strategy could be used to possibly promote a more meniscus-like engineered tissue phenotype [4].“

Minor:

1. TGF-β- The “β” should be in italics. All Greek characters should be in italics.

We would prefer to not make this change – it doesn’t seem to be a standard to always italicize Greek characters. The character isn’t italicized in TGF-β3 in the three PLOS ONE articles we checked – see for example doi: 10.1371/journal.pone.0187349. We also did not use italics for Greek characters in our previous work and so would prefer to leave it as in the original submission. 

2. Please explain: Why concentration of TGF-b3 supplementation was decided as 10 ng/mL?

Thank you for bringing this up. The concentration of 10 ng/mL of TGF-β3 is standard in our research group, originally used for culture of human meniscus cells in 2006 (Adesida, 2006, Arthritis Research & Therapy). It is also a standard concentration (albeit for the isoform TGF-β1 not TGF-β3) used in the chondrogenic medium described originally by Johnstone, 1998, Experimental Cell Research).

We have slightly modified the methods to clarify:

“For tissue formation, cell-seeded scaffolds were cultured in a defined serum-free chondrogenic tissue culture medium, but with or without the standard addition of 10 ng/mL of TGF-β3 in incubators containing 3 or 20% O2 and 5% CO2 (X3 XVivo, Biospherix, USA) for up to 4 weeks(23,24).”

Reviewer #2:

This work appears to be an extension of their previous work. Lacks novelty. It cannot be considered a full paper, excluding the finding they already published, unless they thoroughly revise and expand their studies further. I have uploaded a separate comment sheet.

Comments (from the separate comment sheet): 

The authors seem to have extended their previously published work in the Tissue Engg. Part A. 2019- Hypoxia and TGF-b3 Synergistically Mediate Inner Meniscus-Like Matrix Formation by Fibrochondrocytes

It is good study and fills the gap in our understanding on meniscus matrix-forming ability under dynamic compression. However, since majority of the findings were already established in their previous work, specifically the role of TGF-beta3 in hypoxic conditions backed by good experimental evidence, I feel the current work doesn’t add significant information. It can be a part of the paper but cannot be considered a full paper. I advise authors can consider expanding their study further and resubmit it. Also, they will need to provide a novelty statement in context of their previously published work.

We thank the reviewer for their comments and suggestions. The present study builds upon our previous work: 1. Szojka, 2019, Tissue Engineering Part A; 2. Adesida, 2012, PLOS One; 3. Croutze, 2013, BMC Musculoskeletal Disorders.

We respectfully disagree that the study cannot be considered a full paper. It is in our opinion important that studies such as this one are available to other researchers so that they know what modifications to conditions may be needed.

We also dispute that the present study lacks novelty and thank the reviewer for suggesting inclusion of a novelty statement. The novelty of the study was originally described in the introduction lines 67-73, 81-84, and 85-93. In response to the reviewer’s comments:

1. We now include a summary of previous work in the conclusion before better highlighting the current study’s main results to contextualize the findings – PLOS ONE does not seem to have a dedicated section for a novelty statement but this should have the same effect.

2. We more clearly specify throughout the manuscript the use of continuous hypoxia in the present study (no re-oxygenation during medium changes), which is one difference from the 2012 and 2013 studies using the same type I collagen scaffolds.

3. We provide a complete novelty breakdown of the study below for the reviewer.

Novelty of part 1 (hypoxia and TGF-β3 factorial effects):

Part 1 serves largely to tie up some “loose ends” and eliminate some rival hypotheses related to our previous work, which are described below.

1. The previous work by Szojka (2019) was different in that it used the small-scale pellet model, which is simpler to the type I collagen scaffolds used here. Hypoxia induced a more chondrogenic phenotype in the previous work but not here in part 1. The studies were otherwise similar with the same cell source (meniscus fibrochondrocytes (MFCs) from young donors undergoing partial meniscectomy) and the same continuous hypoxia incubator (Biospherix XVivo). With these variables controlled f¬or, the remaining explanation is that hypoxia affects engineered meniscus differently between the pellet and scaffold models. This is important information for moving forward and motivated much of the discussion (paragraphs 2 & 3).

2. The Adesida and Croutze papers from 2012 & 2013 investigated hypoxia in the larger-scale type I collagen scaffold, with three important differences to the present work (besides having different objectives): 1. They used MFCs from donors of advanced age undergoing total meniscus removal for severe osteoarthritis compared to MFCs from young donors undergoing partial meniscectomy for (comparatively) acute injuries here, and 2. these studies used hypoxia incubators that required re-oxygenation of tissues during media changes, whereas the present study used the XVivo incubator that eliminated re-oxygenation as media was changed within a 3% atmosphere environment. We had supposed that continuous hypoxia without reoxygenation could possibly induce chondrogenic differentiation of the MFCs in the type I collagen scaffold without supplemented growth factors (as was observed in Marsano, 2016 in human bone marrow-derived mesenchymal stem cells). 3. These previous studies had hypoxia & normoxia maintained during expansion culture prior to seeding the scaffolds, whereas the present work had pure normoxia expansion prior to division into normoxia and hypoxia for scaffold culture.

Yet, hypoxia had similar limited effects on the MFC matrix-forming phenotype in the older studies to the present study. Hypoxia also had limited effects in the present study when TGF-β3 was not supplemented in the medium. This result shows that the reason for limited hypoxia effects on MFC matrix-forming phenotype in the type I collagen scaffold is not the donor source, reoxygenation, or oxygen tension during cell expansion. This motivated much of the discussion in the present work and led us to suggest that the hypoxia treatment in the type I collagen scaffold may need to be timed to be after a pre-culture period in normoxia due to oxygen effects on scaffold contraction. We have recent data in support of this conclusion. It is from a separate project and so we are reluctant to include it in the present manuscript, although we are happy to reproduce it below for the reviewer.

Novelty of part 2 of the present study (dynamic compression and TGF-β3 factorial effects under hypoxia)

Part 2 is the most novel part of the present study. It has the following unique features:

1. To our knowledge, no published work investigates dynamic compression effects on matrix biosynthesis and gene expression in human meniscus, in either explants or engineered tissues.

2. Further, very few studies combine dynamic compression with hypoxia in any culture model. Research is limited in this field even though it Is perhaps the most relevant for inner meniscus and cartilage.

3. It was not known prior to this study whether hypoxia alone or hypoxia combined with dynamic compression of MFCs in the type I collagen scaffold as applied would be sufficient to induce fibrocartilage matrix formation without growth factor supplementation. Hypoxia alone was sufficient to induce chondrogenic differentiation of human MSCs (Marsano, 2016, Stem Cells Translational Medicine). However, hypoxia and hypoxia+ DC were clearly not sufficient based on our results, which is an important result. Growth factors or modified experimental conditions such as a more aggressive loading regime may be necessary to induce matrix formation.

---

## [Decision Letter · Decision Letter 1]

17 Feb 2021

PONE-D-20-33847R1

Engineered human meniscus' matrix-forming phenotype is unaffected by low strain dynamic compression under hypoxic conditions

PLOS ONE

Dear Dr. Adesida,

Thank you for submitting your manuscript to PLOS ONE. After careful consideration, we feel that it has merit but does not fully meet PLOS ONE’s publication criteria as it currently stands. Therefore, we invite you to submit a revised version of the manuscript that addresses the points raised during the review process.

We look forward to receiving your revised manuscript.

Kind regards,

Feng Zhao

Academic Editor

PLOS ONE

**Comments to the Author**

Reviewer #2: The authors claim in the conclusion section seems incorrect. Line 438-442: The present study demonstrates that, in a type I collagen scaffold seeded with MFCs from partial meniscectomy, neither continuous hypoxia under static conditions nor hypoxia combined with dynamic compression to 10% strain induced a fibrocartilaginous inner meniscus-like matrix-forming phenotype, regardless of TGF-beta supplementation.- From their data, TGF beta is indeed an important factor to to induce fibrocartilaginous matrix formation! They need to clarify.

---

## [Author Response · Author response to Decision Letter 1]

17 Feb 2021

We thank the reviewers and editor for their time in reviewing our work and address the outstanding comment below.

Reviewer #2: The authors’ claim in the conclusion section seems incorrect. Line 438-442: The present study demonstrates that in a type I collagen scaffold seeded with MFCs from partial meniscectomy, neither continuous hypoxia under static conditions nor hypoxia combined with dynamic compression to 10% strain induced a fibrocartilaginous inner meniscus-like matrix-forming phenotype, regardless of TGF-beta supplementation. From their data, TGF beta is indeed an important factor to induce fibrocartilaginous matrix formation! They need to clarify.

Thank you for identifying this problem with our original text. We realize now that the original writing (“regardless of TGF-β3 supplementation”) failed to make the distinction below:

• Hypoxia and hypoxia with dynamic compression did not induce a fibrocartilaginous phenotype.

• Hypoxia and hypoxia with dynamic compression did not promote the fibrocartilaginous phenotype when it was induced by TGF-β3 supplementation

Thus, we have now clarified the concluding text:

The present study demonstrates that in a type I collagen scaffold seeded with MFCs from partial meniscectomy, neither continuous hypoxia under static conditions nor hypoxia combined with dynamic compression to 10% strain induced a fibrocartilaginous inner meniscus-like matrix-forming phenotype in the absence of TGF-β3 supplementation. They also did not promote the inner meniscus-like matrix-forming phenotype when it was induced by TGF-β3 supplementation.

---

## [Editor Report · Decision Letter 2]

24 Feb 2021

Engineered human meniscus' matrix-forming phenotype is unaffected by low strain dynamic compression under hypoxic conditions

PONE-D-20-33847R2

Dear Dr. Adesida,

We’re pleased to inform you that your manuscript has been judged scientifically suitable for publication and will be formally accepted for publication once it meets all outstanding technical requirements.

Kind regards,

Feng Zhao

Academic Editor

PLOS ONE

---

## [Editor Report · Acceptance letter]

1 Mar 2021

PONE-D-20-33847R2 

Engineered human meniscus' matrix-forming phenotype is unaffected by low strain dynamic compression under hypoxic conditions 

Dear Dr. Adesida:

I'm pleased to inform you that your manuscript has been deemed suitable for publication in PLOS ONE. Congratulations! Your manuscript is now with our production department. 

Kind regards, 

on behalf of

Dr. Feng Zhao 

Academic Editor

PLOS ONE